# Consistency Verification for Detecting AI-Generated Images

## Abstract

With the rapid development of generative models, AI-generated images have sparked significant concerns regarding their potential misuse for malicious purposes, highlighting the urgent need for AI-generated image detection. Current methods primarily focus on training a binary classifier to detect generated images. However, the efficacy of these methods is critically dependent on the quantity and quality of the collected AI-generated images. More importantly, they suffer from a generalization challenge: *the literature lacks sufficient exploration of whether a binary classifier trained on images from a specific diffusion model can effectively generalize to images generated by other models.* In this work, we propose a novel framework termed **con**sistency **v**erification (ConV) for AI-generated image detection, providing a new approach that detects without requiring AI-generated images. In particular, we introduce two functions and establish a principle for designing them so that their outputs remain consistent for natural images but exhibit significant inconsistency for AI-generated images. Our principle shows that gradients of these two functions need to lie within two mutually orthogonal subspaces. This enables a training-free detection approach: an image is identified as AI-generated if transformation along its data manifold results in a substantial change in the loss value of a self-supervised model pre-trained on natural images. This detection framework leads to the unique advantage of ConV over existing methods: *ConV identifies AI-generated images by fitting the distribution of natural images rather than that of AI-generated images.* Extensive experiments across various benchmarks validate the effectiveness of the proposed ConV.

## 1 INTRODUCTION

Recent advances in generative models have revolutionized image generation, making it possible to create highly realistic images (Rombach et al., 2022; Dhariwal & Nichol, 2021; Karras et al., 2019). While these generative models offer impressive capabilities, they also introduce significant risks, including the proliferation of deepfakes and other manipulated content. The realism achieved by these technologies raises urgent concerns about their potential misuse in sensitive areas like politics and economics. Moreover, if we simply use AI-generated images as part of the training data, the trained model may largely degrade its quality Shumailov et al. (2024), so it is essential to distinguish between natural images and AI-generated ones. To deal with these potentially dire risks, various AI-generated image detection methods have been developed. In this regard, a common approach is to consider generated image detection as a binary classification task. To train a binary classifier for detecting generated images, current methods typically require to collect numerous natural and generated images to construct a training dataset (Chai et al., 2020; Wang et al., 2020).

Although current methods have achieved exciting success, they often struggle to generalize well to images generated by unknown generative models. To promote the generalization ability on images generated by unknown generative models, one possible approach is to construct a more extensive training dataset by collecting more natural and generated images for training the binary classifier (Jeong et al., 2022; Tan et al., 2024). Besides collecting data, advanced methods propose to introduce pre-trained models as priors to promote the generalization ability. Some works, inspired by the recent success of large models, propose to detect generated images by leveraging features extracted by these large models (Ojha et al., 2023; Liu et al., 2024), such as CLIP (Radford et al., 2021). Meanwhile, some works propose to leverage the reconstruction capabilities of pre-trained diffusion

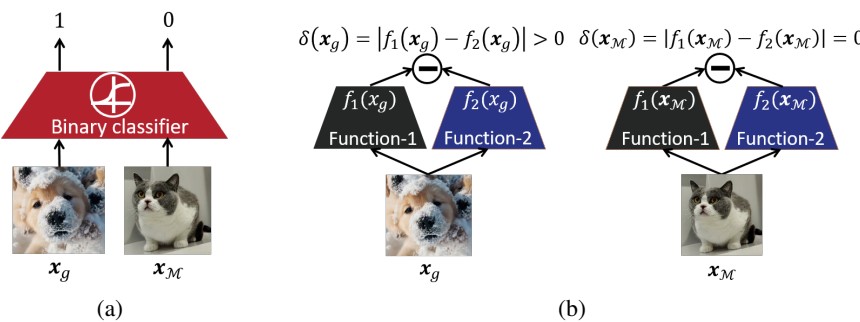

Figure 1: Comparison of (a): the existing framework, and (b): our proposed ConV. The binary classifier in (a) is trained using natural images $\mathbf{x}_{\mathcal{M}}$ and AI-generated images $\mathbf{x}_g$, thereby, its efficacy relies on both the natural and generated data distributions. In contrast, the two functions in (b) are trained on natural data distribution, leading to the advantage of ConV: identifying AI-generated images by fitting the distribution of natural images rather than that of AI-generated images.

models (Wang et al., 2023a; Ricker et al., 2024). Although these methods have achieved outstanding results, they require a lot of natural and generated images to train a binary classifier, making the current methods computationally intensive. Moreover, sustaining robust detection performance necessitates the continual collection of images generated by the latest generative models, which can be costly or even infeasible due to the inaccessibility of potential models, e.g., Sora OpenAI (2024).

Hence, the major challenge for the existing methods is ensuring that the binary classifier generalizes effectively across diverse unknown generative models. This stems from the fact that these binary classifiers are trained over natural and generated images to distinguish between these two types of images. Thus, the performance of these binary classifiers relies on the diversity of generated data. Unfortunately, it is challenging to determine whether a binary classifier trained over images generated by some diffusion models can generalize to those generated by other models. Their defects of heavy dependence on generated image distribution underscore the necessity of exploring a novel framework for generated image detection, where the detector's performance relies on the natural data distribution rather than the generated image distribution. However, this remains challenging, because the literature has yet to determine whether models training merely on natural images can be leveraged to distinguish between natural and generated images effectively, and if yes, how and why?

To address the challenge, we propose a novel framework for detecting generated images called **con**sistency **v**erification (ConV). As shown in Figure 1, we introduce two functions, aiming to detect generated images by ensuring that the outputs of these functions remain consistent for natural images but exhibit significant inconsistency for generated images. To this end, we establish a principle (see Eq. 6) to design these functions based on our theoretical analysis: outputs of these two functions are the same on the natural distribution while their gradients need to lie within two mutually orthogonal subspaces. This enables a training-free detection approach (see Eq. 12): if an image transformed along its data manifold induces a substantial change in the loss value of a self-supervised model pre-trained over natural images, it is identified as generated. The advantage of ConV over existing methods is its reliance on fitting the natural data distribution rather than the distribution of generated images. Comprehensive experiments across various benchmarks for generated image detection demonstrate the effectiveness of the proposed ConV (see Tables 1-3). To further verify the effectiveness of the proposed ConV, we collect images generated by Sora OpenAI (2024) and OpenSora Zheng et al. (2024) and compare ConV with baselines. The experiments demonstrate the efficacy and robustness of ConV against variations in generative models (see Table 4).

We summarize our main contributions as follows:

- We highlight the generalization issue of existing works: it is challenging to determine whether a detector trained over images generated by some diffusion models can generalize to those generated by other models. This motivates a promising direction to explore detectors whose detection ability relies solely on fitting the natural data distribution.

- We propose a novel framework for detecting AI-generated images called **con**sistency **v**erification (ConV), which detects images by verifying the consistency of two functions. The design of these two functions is guided by our orthogonality principle. Namely, gradients of these functions need to lie within two mutually orthogonal subspaces (Eq 6).

- Our proposed orthogonality principle enables a training-free approach to detecting AI-generated images by leveraging the consistency of a pre-trained self-supervised model on images before and after perturbations along the data manifold. Extensive experiments conducted on various standard benchmarks and datasets collected from Sora demonstrate the effectiveness and robustness of the proposed method (Tables 1-4).

## 2 CONSISTENCY VERIFICATION

In this section, we will give the motivation 2.1, objective 2.2, and realization 2.3 of consistency verification proposed for AI-generated image detection.

### 2.1 MOTIVATION

Humans can distinguish AI-generated images from natural images through some types of indescribable differences in patterns. Intuitively, humans know that if a natural image captures the same content as a given AI-generated image, the natural image will be different. In contrast, if we degrade natural images along its data manifold, e.g., tiny affine transformation, the degraded natural images are still discriminated as natural images.

To formally characterize this discrepancy, we present the following notations. Let $\mathbf{x} \in \mathcal{X} \subset \mathbb{R}^d$ denote the image, where $d$ denotes the dimension of images. To distinguish, we use $\mathbf{x}_n$ and $\mathbf{x}_g$ to denote the natural and AI-generated image. In particular, for a given generated image $\mathbf{x}_g$, even if it captures similar content to a natural image $\mathbf{x}_n$, humans know they are distinguishable in certain ways. This can be formulated by projecting the generated image $\mathbf{x}_g$ onto the point $\mathbf{x}_{\mathcal{M}(\mathbf{x}_g)}$ on the data manifold $\mathcal{M}$, i.e.,

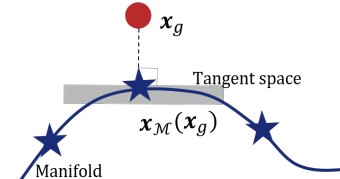

Figure 2: Illustration of projecting a generated image $\mathbf{x}_g$ onto the data manifold $\mathcal{M}$.

$$\mathbf{x}_{\mathcal{M}}(\mathbf{x}_g) = \arg \min_{\mathbf{x}' \in \mathcal{M}} d(\mathbf{x}', \mathbf{x}_g), \ \mathbf{x}_{\mathcal{M}}(\mathbf{x}_g) \in \mathcal{M}, \ \mathbf{x}_g \notin \mathcal{M}, \tag{1}$$

where $\mathbf{x}_{\mathcal{M}}(\mathbf{x}_g)$ stands for the point closest to $\mathbf{x}_g$ on the data manifold of natural images $\mathcal{M}$ and $d$ is a metric. Namely, images on the data manifold $\mathcal{M}$ are considered natural, whereas those deviating from $\mathcal{M}$ are regarded as AI-generated.

In this context, the data manifold perspective provides an intuitive framework for understanding the difference. In particular, transforming natural image $\mathbf{x}_{\mathcal{M}}$ along the local tangent space $\mathcal{T}(\mathbf{x}_{\mathcal{M}})$, leading to the fact that the degraded images are still on the data manifold. In contrast, even the discrepancy $d(\mathbf{x}_{\mathcal{M}}(\mathbf{x}_g), \mathbf{x}_g)$ is minimal, $\mathbf{x}_g$ is considered as generated, because $\mathbf{x}_g$ departures from the manifold. Intuitively, even a slight discrepancy between $\mathbf{x}_{\mathcal{M}}(\mathbf{x}_g)$ and $\mathbf{x}_g$ allows us to identify the difference between a generated image and the corresponding natural image on the data manifold. Thus, we consider the discrepancy between a generated image and its closest natural image on the data manifold to represent the direction of the fastest departure from the manifold. This means that

$$\mathbf{v}^\top (\mathbf{x}_{\mathcal{M}}(\mathbf{x}_g) - \mathbf{x}_g) = 0, \ \mathbf{v} \in \mathcal{T}(\mathbf{x}_{\mathcal{M}}(\mathbf{x}_g)). \tag{2}$$

This discrepancy inspires us to introduce two functions to detect generated images, where these two functions are related to the tangent space and the space orthogonal to the tangent space, respectively.

### 2.2 OBJECTIVE

Aligning with the motivation, we introduce a two-function framework for generated image detection. In particular, we propose a consistency verification framework where the two introduced functions are devised to be consistent over natural images and inconsistent over AI-generated images. Namely,

this framework detects generated images by verifying the consistency of the two functions. Specifically, let $f_1(\cdot) : \mathbb{R}^d \to \mathbb{R}$ and $f_2(\cdot) : \mathbb{R}^d \to \mathbb{R}$ be the two functions. Then, the inconsistency $|f_1(\cdot) - f_2(\cdot)|$ between these two functions can be employed to detect generated images. Namely, generated images can be detected by $\mathbb{I}(|f_1(\cdot) - f_2(\cdot)| > \alpha)$ with the threshold $\alpha$.

For images on the manifold, we make these two functions consistent by setting

$$\delta(\mathbf{x}_\mathcal{M}) = |f_1(\mathbf{x}_\mathcal{M}) - f_2(\mathbf{x}_\mathcal{M})| = 0, \tag{3}$$

where we denote $\mathbf{x}_\mathcal{M}(\mathbf{x}_g)$ as $\mathbf{x}_\mathcal{M}$ for simplicity. Then, the objective is to devise the two functions to ensure that the inconsistency over generated images is larger than that over the natural images, i.e., $\delta(\mathbf{x}_g) \geq \delta(\mathbf{x}_\mathcal{M})$. In this regard, we show that (with more details in the appendix)

$$\delta(\mathbf{x}_g) \geq |\,|\nabla f_1(\mathbf{x}_\mathcal{M})^\top (\mathbf{x}_g - \mathbf{x}_\mathcal{M})| - |\nabla f_2(\mathbf{x}_\mathcal{M})^\top (\mathbf{x}_g - \mathbf{x}_\mathcal{M})|\,| \geq 0 = \delta(\mathbf{x}_\mathcal{M}), \tag{4}$$

where equality holds if, and only if, the absolute values of the two quantities are identical. According to Eq. 4, enlarging the difference between these two terms, i.e., $|\nabla f_1(\mathbf{x}_\mathcal{M})^\top (\mathbf{x}_g - \mathbf{x}_\mathcal{M})|$ and $|\nabla f_2(\mathbf{x}_\mathcal{M})^\top (\mathbf{x}_g - \mathbf{x}_\mathcal{M})|$ will make the natural and generated images separable. Thus, the objective of consistency verification is to maximize one term and minimize the other term while keeping the output values of these two functions the same. This can be formalized by

$$\min_{f_1, f_2 \in \mathcal{F}} |\nabla f_1(\mathbf{x}_\mathcal{M})^\top (\mathbf{x}_g - \mathbf{x}_\mathcal{M})| - |\nabla f_2(\mathbf{x}_\mathcal{M})^\top (\mathbf{x}_g - \mathbf{x}_\mathcal{M})|, \text{s.t. } f_1(\mathbf{x}_\mathcal{M}) = f_2(\mathbf{x}_\mathcal{M}), \tag{5}$$

where $\mathcal{F}$ denotes a hypothesis space.

However, learning these two functions using Eq. 5 still relies on the generated data, i.e., $\mathbf{x}_g$. To decouple function optimization from the generated data distribution, we leverage orthogonality priors from the motivation 2.1 to provide design principles for these functions. According to the above discussion, one straightforward approach to realizing $f_1$ and $f_2$ is to devise these functions such that their gradients for the input lie in two orthogonal subspaces, i.e., the tangent space and the space orthogonal to the tangent space. This orthogonality principle can be formalized as,

$$\nabla f_1(\mathbf{x}_\mathcal{M}) \in \mathcal{O}(\mathbf{x}_\mathcal{M}), \ \nabla f_2(\mathbf{x}_\mathcal{M}) \in \mathcal{T}(\mathbf{x}_\mathcal{M}), \ f_1(\mathbf{x}_\mathcal{M}) = f_2(\mathbf{x}_\mathcal{M}) \tag{6}$$

where $\mathcal{O}(\mathbf{x}_\mathcal{M})$ denotes the subspace orthogonal to the tangent space $\mathcal{T}(\mathbf{x}_\mathcal{M})$. Then, we have

$$\delta(\mathbf{x}_g) \geq ||\nabla f_1(\mathbf{x}_\mathcal{M})^\top \mathbf{p}| - |\nabla f_2(\mathbf{x}_\mathcal{M})^\top \mathbf{p}|| = |\nabla f_1(\mathbf{x}_\mathcal{M})^\top \mathbf{p}| > 0 = \delta(\mathbf{x}_\mathcal{M}), \tag{7}$$

where $\mathbf{p} = \mathbf{x}_g - \mathbf{x}_\mathcal{M}$ denotes the difference between a generated image and its corresponding point on the data manifold, the equation holds due to the conclusion in Eq. 2, and the inequality holds because the probability that two vectors in the same space are orthogonal is zero. Consequently, the orthogonality principle ensures that these two functions are consistent on natural images, i.e., $f_1(\mathbf{x}_\mathcal{M}) = f_2(\mathbf{x}_\mathcal{M})$, while inconsistent on generated images, i.e., $|\delta(\mathbf{x}_g)| > |\delta(\mathbf{x}_\mathcal{M})| = 0$.

### 2.3 REALIZATION

In this work, we propose a training-free approach to construct these two functions. The reason is twofold: i) our framework allows the training-free construction of these functions, and ii) we aim to validate the effectiveness of the orthogonality principle without incurring significant energy costs, as fitting the distribution of natural data requires a lot of data and computing power for training.

It shows that well-trained models are typically insensitive to the transformation along the data manifold Simard et al. (1991); Bengio et al. (2013); Rifai et al. (2011). This can be formalized as,

$$(\mathbf{v} - \mathbf{x}_\mathcal{M})^\top \frac{\partial \ell(\mathbf{x}_\mathcal{M})}{\partial \mathbf{x}_\mathcal{M}} \approx 0, \quad \mathbf{v} \in \mathcal{T}(\mathbf{x}_\mathcal{M}), \tag{8}$$

where $\mathbf{v}$ stands for the point sampled from the tangent space $\mathcal{T}(\mathbf{x}_\mathcal{M})$ and $\ell(\cdot)$ is the loss function of a model. This implies that $\frac{\partial \ell(\mathbf{x}_\mathcal{M})}{\partial \mathbf{x}_\mathcal{M}}$ is orthogonal to the tangent space $\mathcal{T}(\mathbf{x}_\mathcal{M})$, which is consistent with the direction $\mathbf{p} = \mathbf{x}_g - \mathbf{x}_\mathcal{M}$, as shown in Eq 2. Hence, we propose to realize $f_1(\cdot)$ using a well-trained neural network. This means that both $\frac{\partial \ell(\mathbf{x}_\mathcal{M})}{\partial \mathbf{x}_\mathcal{M}}$ and $\mathbf{p}$ lies in the subspace orthogonal to tangent space $\mathcal{T}(\mathbf{x}_\mathcal{M})$. This is consistent with the principle, i.e., $\nabla f_1(\mathbf{x}_\mathcal{M}) \in \mathcal{O}(\mathbf{x})$. We have

$$|\nabla f_1(\mathbf{x}_\mathcal{M})^\top \mathbf{p}| = |\frac{\partial \ell(\mathbf{x}_\mathcal{M})}{\partial \mathbf{x}_\mathcal{M}}^\top \mathbf{p}| = \left\|\frac{\partial \ell(\mathbf{x}_\mathcal{M})}{\partial \mathbf{x}_\mathcal{M}}\right\| \|\mathbf{p}\| |\cos(\frac{\partial \ell(\mathbf{x}_\mathcal{M})}{\partial \mathbf{x}_\mathcal{M}}, \mathbf{p})| > 0, \tag{9}$$

where $\mathbf{p} = \mathbf{x}_g - \mathbf{x}_{\mathcal{M}}$ is the difference between natural and generated images, and the last inequality holds because the probability that two vectors in the same space are orthogonal is zero. We propose to realize $f_1(\cdot)$ using models trained with self-supervised learning, which would avoid the reliance on image labels used in classification tasks. This is because obtaining the loss value of a classification model requires labels that could be hard to obtain in many practical scenarios.

For the second term, we will realize it using the orthogonality such that $\nabla f_2(\mathbf{x}_{\mathcal{M}}) \in \mathcal{T}(\mathbf{x}_{\mathcal{M}})$ or $|\nabla f_2(\mathbf{x}_{\mathcal{M}})^\top \mathbf{p}| = 0$. We achieve this by introducing the local tangent space into $\nabla f_2(\mathbf{x})$. To this end, we propose to realize $f_2$ using a composite function: $f_2 := f_1 \circ h$. This leads to the fact that

$$\nabla f_2(\mathbf{x}_{\mathcal{M}}) = \mathbf{J}_{h(\mathbf{x}_{\mathcal{M}})} \frac{\partial f_1(h(\mathbf{x}_{\mathcal{M}}))}{\partial h(\mathbf{x}_{\mathcal{M}})}, \tag{10}$$

where $\mathbf{J}_{h(\mathbf{x}_{\mathcal{M}})}$ is the Jacobian matrix of the function $h(\mathbf{x}_{\mathcal{M}})$. If $h(\cdot)$ models the transformation along local data manifold, $\mathbf{J}_{h(\mathbf{x}_{\mathcal{M}})}$ models the tangent space at point $\mathbf{x}_{\mathcal{M}}$. Then, we have

$$\nabla f_2(\mathbf{x}_{\mathcal{M}})^\top \mathbf{p} = \frac{\partial f_1(h(\mathbf{x}_{\mathcal{M}}))}{\partial h(\mathbf{x}_{\mathcal{M}})}^\top \mathbf{J}_{h(\mathbf{x}_{\mathcal{M}})}^\top \mathbf{p} = 0, \tag{11}$$

where $\mathbf{J}_{h(\mathbf{x})}^\top$ denotes the tangent space orthogonal to the vector $\mathbf{p} = \mathbf{x}_g - \mathbf{x}_{\mathcal{M}}$, see Eq. 2.

For the last term in the orthogonality principle, we should ensure that $f_1(\mathbf{x}_{\mathcal{M}}) = f_2(\mathbf{x}_{\mathcal{M}}) := f_1(h(\mathbf{x}_{\mathcal{M}}))$. There are numerous approaches to realize $h(\cdot)$. In this regard, we propose to leverage data transformation functions used in the training phase to realize $h(\cdot)$, because self-supervised models are trained to be insensitive to these transformations along local data manifold under various self-supervised learning scenarios (Yu et al., 2023; Jaderberg et al., 2015). Thus, for a given input image $\mathbf{x}$, we can determine whether it is generated by calculating the consistency $\delta(\mathbf{x})$,

$$\delta(\mathbf{x}) = |f_1(\mathbf{x}) - f_1(h(\mathbf{x}))| \begin{cases} = 0, & \mathbf{x} \in \mathcal{M}, \\ > 0, & \mathbf{x} \notin \mathcal{M}. \end{cases} \tag{12}$$

Technically, our training-free realization is equal to verifying the robustness of a pre-trained self-supervised model $f_1(\cdot)$ against the data transformations $h(\cdot)$. Here, $f_1(\cdot)$ merely fits the natural data distribution, avoiding the reliance on the distribution of AI-generated images.

## 2.4 OVERVIEW

An overview of the proposed consistency verification is presented in Figure 3. As shown in the figure, our method is training-free and seamlessly deployed in practical scenarios. Specifically, we merely download a neural network pre-trained with a self-supervised learning task over a large-scale dataset.

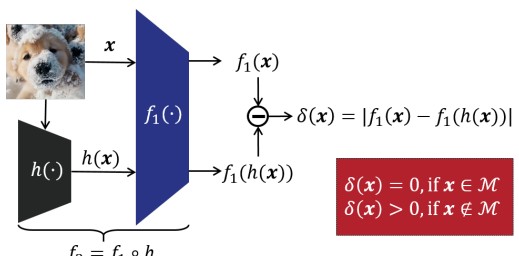

Figure 3: Framework of the proposed consistency verification.

Subsequently, we obtain the loss values of both the original and transformed images. Ultimately, images are identified as generated if the difference between loss values exceeds a predetermined threshold. We can apply multiple random transformations and compute corresponding loss function values if computational resources allow. Intuitively, this would result in more accurate detection performance, which is fortunately consistent with our experiments, see Figure 4.

Negative samples are widely used in self-supervised learning, which could increase the computational cost of generated image detection. Inspired by a recent work Oquab et al. (2024), we calculate the similarity of representation $\mathbf{r} = \phi(\mathbf{x})$, where $\phi(\cdot)$ is the feature extractor of a self-supervised model. The feasibility results from the objective function used in self-supervised learning,

$$\log P(\mathbf{x}) = \log \frac{e^{(\mathbf{r}^\top \mathbf{r}_h / \tau)}}{\sum_{\mathbf{z}_-} e^{(\mathbf{r}^\top \mathbf{r}_- / \tau)} + e^{(\mathbf{r}^\top \mathbf{r}_h / \tau)}} = \log \frac{1}{\sum_{\mathbf{z}_-} e^{(\mathbf{r}^\top \mathbf{r}_- / \tau) - (\mathbf{r}^\top \mathbf{r}_h / \tau)} + 1}, \tag{13}$$

Table 1: AI-generated image detection performance on ImageNet. Values are percentages. **Bold** numbers are superior results and the underlined italicized values are the second-best performance.

| Methods | ADM | | ADMG | | LDM | | DiT | | BigGAN | | GigaGAN | | StyleGAN XL | | RQ-Transformer | | Mask GIT | | Average | |
|---|---|---|---|---|---|---|---|---|---|---|---|---|---|---|---|---|---|---|---|---|
| | AUROC | AP | AUROC | AP | AUROC | AP | AUROC | AP | AUROC | AP | AUROC | AP | AUROC | AP | AUROC | AP | AUROC | AP | AUROC (↑) | AP (↑) |
| *Training-based Methods* | | | | | | | | | | | | | | | | | | | | |
| CNNspot | 62.25 | 63.13 | 63.28 | 62.27 | 63.16 | 64.81 | 62.85 | 61.16 | 85.71 | 84.93 | 74.85 | 71.45 | 68.41 | 68.67 | 61.83 | 62.91 | 60.98 | 61.69 | 67.04 | 66.78 |
| Ojha | 83.37 | 82.95 | 79.60 | 78.15 | 80.35 | 79.71 | 82.93 | 81.72 | **93.07** | **92.77** | 87.45 | 84.88 | 85.36 | 83.15 | 85.19 | 84.22 | **90.82** | **90.71** | 85.35 | 84.25 |
| DIRE | 51.82 | 50.29 | 53.14 | 52.96 | 52.83 | 51.84 | 54.67 | 55.10 | 51.62 | 50.83 | 50.70 | 50.27 | 50.95 | 51.36 | 55.95 | 54.83 | 52.58 | 52.10 | 52.70 | 52.18 |
| NPR | 85.68 | 80.86 | **84.34** | 79.79 | **91.98** | **86.96** | **86.15** | 81.26 | 89.73 | 84.46 | 82.21 | 78.20 | 84.13 | 78.73 | 80.21 | 73.21 | 89.61 | 84.15 | 86.00 | 80.84 |
| *Training-free Methods* | | | | | | | | | | | | | | | | | | | | |
| AEROBLADA | 55.61 | 54.26 | 61.57 | 56.58 | 62.67 | 60.93 | 85.88 | 87.71 | 44.36 | 45.66 | 47.39 | 48.14 | 47.28 | 48.54 | 67.05 | 67.69 | 48.05 | 48.75 | 57.87 | 57.85 |
| RIGID | 87.00 | 85.29 | 81.22 | 77.90 | 74.60 | 69.51 | 70.22 | 67.17 | 87.81 | 86.23 | 85.54 | 84.39 | 86.58 | 86.41 | 90.66 | 89.89 | 89.94 | 88.41 | 83.73 | 81.69 |
| ConV | **88.89** | **86.60** | 82.46 | 79.83 | 78.94 | 75.88 | 75.25 | 70.11 | 92.83 | 92.05 | **91.89** | **90.93** | **92.15** | **91.82** | **93.02** | **91.26** | 88.79 | 87.88 | **87.13** | **85.15** |

Table 2: AI-generated image detection performance on LSUN-BEDROOM. **Bold** numbers are superior results and the underlined italicized values are the second-best performance.

| Methods | ADM | | DDPM | | iDDPM | | Diffusion GAN | | Projected GAN | | StyleGAN | | Unleashing Transformer | | Average | |
|---|---|---|---|---|---|---|---|---|---|---|---|---|---|---|---|---|
| | AUROC | AP | AUROC | AP | AUROC | AP | AUROC | AP | AUROC | AP | AUROC | AP | AUROC | AP | AUROC (↑) | AP (↑) |
| *Training-based Methods* | | | | | | | | | | | | | | | | |
| CNNspot | 64.83 | 64.24 | 79.04 | 80.58 | 76.95 | 76.28 | 88.45 | 87.19 | 90.80 | 89.94 | **95.17** | **94.94** | 93.42 | 93.11 | 84.09 | 83.75 |
| Ojha | 71.26 | 70.95 | 79.26 | 78.27 | 74.80 | 73.46 | 84.56 | 82.91 | 82.00 | 78.42 | 81.22 | 78.08 | 83.58 | 83.48 | 79.53 | 77.94 |
| DIRE | 57.19 | 56.85 | 61.91 | 61.35 | 59.82 | 58.29 | 53.18 | 53.48 | 55.35 | 54.93 | 57.66 | 56.90 | 67.92 | 68.33 | 59.00 | 58.59 |
| NPR | **75.43** | **72.60** | 91.42 | 90.89 | 89.49 | 88.25 | 76.17 | 74.19 | 75.07 | 74.59 | 68.82 | 63.53 | 84.39 | 83.67 | 80.11 | 78.25 |
| *Training-free Methods* | | | | | | | | | | | | | | | | |
| AEROBLADA | 57.05 | 58.37 | 61.57 | 61.49 | 59.82 | 61.06 | 47.12 | 48.25 | 45.98 | 46.15 | 45.63 | 47.06 | 59.71 | 57.34 | 53.85 | 54.25 |
| RIGID | 69.76 | 68.31 | 88.35 | 88.82 | 84.15 | 84.54 | 91.85 | 92.28 | 92.65 | 93.18 | 78.09 | 76.54 | 91.94 | 92.28 | 85.25 | 85.13 |
| ConV | 73.71 | 71.52 | 87.74 | 86.59 | 82.96 | 81.79 | **93.79** | **93.87** | **94.73** | **94.74** | 84.10 | 82.35 | **93.75** | **93.51** | **87.25** | **86.34** |

where $\mathbf{r}_h$ is the representation of $h(\mathbf{x})$ and $\mathbf{r}_-$ denotes the representation of negative samples. Thus, we can employ the similarity between representations, i.e., $\mathbf{r}^\top \mathbf{r}_h$, as a surrogate of loss value. This avoids the use of negative samples. Note that applying a softmax function to the representation $\mathbf{r}$ leads to the objective function used in previous works Caron et al. (2021); Oquab et al. (2024). In this context, the high similarity between the representation of images and transformed images means the consistency between functions, i.e., detected as natural images.

## 3 EXPERIMENTS

This section aims to verify the effectiveness of the proposed ConV, especially for practical scenarios with unknown generative models. Before that, we will detail the experimental setups.

### 3.1 EXPERIMENT SETUP

**Datasets and generative models.** We evaluate the performance of ConV and baseline methods on widely used benchmarks: ImageNet (Deng et al., 2009) and LSUN-BEDROOM (Yu et al., 2015) with generated images provided by (Stein et al., 2023). For ImageNet, fake images are generated with ADM (Dhariwal & Nichol, 2021), ADM-G, LDM (Rombach et al., 2022), DiT-XL2 (Peebles & Xie, 2023), BigGAN (Brock et al., 2019), GigaGAN (Kang et al., 2023), StyleGAN (Karras et al., 2019), RQ-Transformer (Lee et al., 2022), and MaskGIT (Chang et al., 2022). For LSUN-BEDROOM, fake images are generated with ADM, DDPM (Ho et al., 2020), iDDPM (Nichol & Dhariwal, 2021), Diffusion Projected GAN (Wang et al., 2023b), Projected GAN (Wang et al., 2023b), StyleGAN (Karras et al., 2019) and Unleasing Transformer (Bond-Taylor et al., 2022). We further evaluate methods using GenImage Dataset (Zhu et al., 2023), which primarily employs diffusion models for image generation with generators including Stable Diffusion V1.4 (Rombach et al., 2022), Stable Diffusion V1.5 (Rombach et al., 2022), GLIDE, VQDM (Gu et al., 2022), Wukong (Wukong), BigGAN, ADM, and Midjourney (Midjourney). This dataset contains $1,331,167$ natural and $1,350,000$ AI-generated images.

Current advancements in generative technology have significantly enhanced the realism of synthetic videos (Khachatryan et al., 2023; Blattmann et al., 2023), thereby raising substantial concerns regarding trust in digital media. Moreover, the inaccessibility of their parameters and even their architectures underscores the necessity of verifying the generalization capability of newly proposed detection methods over these generative models. To verify whether the proposed ConV generalizes to these challenging scenarios, we download videos generated by these models and detect images sampled from these videos. Since we currently cannot access the generative model used in Sora (OpenAI, 2024), we gathered several publicly available videos and extracted $1,000$ frames. Additionally, we generate 100 videos through the open-source OpenSora project (Zheng et al., 2024), extracting

Table 3: AI-generated image detection performance on GenImage. Except for ConV and RIGID, all methods require training on images generated by SD V1.4.

| Methods | Midjourney | SD V1.5 | ADM | GLIDE | Wukong | VQDM | BigGAN | Avg ACC(%) |
|---|---|---|---|---|---|---|---|---|
| | | | | | Models | | | |
| | | | | Training-based Methods | | | | |
| ResNet-50 | 54.90 | 99.70 | 53.50 | 61.90 | 98.20 | 56.60 | 52.00 | 68.11 |
| DeiT-S | 55.60 | 99.80 | 49.80 | 58.10 | 98.90 | 56.90 | 53.50 | 67.51 |
| Swin-T | 62.10 | 99.80 | 49.80 | 67.60 | 99.10 | 62.30 | 57.60 | 71.19 |
| CNNspot | 52.80 | 95.90 | 50.10 | 39.80 | 78.60 | 53.40 | 46.80 | 58.63 |
| Spec | 52.00 | 99.20 | 49.70 | 49.80 | 94.80 | 55.60 | 49.80 | 64.41 |
| F3Net | 50.10 | **99.90** | 49.90 | 50.00 | **99.90** | 49.90 | 49.90 | 64.22 |
| GramNet | 54.20 | 99.10 | 50.30 | 54.60 | 98.90 | 50.80 | 51.70 | 65.66 |
| DIRE | 60.20 | 99.80 | 50.90 | 55.00 | 99.20 | 50.10 | 50.20 | 66.49 |
| Ojha | 73.20 | 84.00 | 55.20 | 76.90 | 75.60 | 56.90 | 80.30 | 71.73 |
| LaRE | 66.40 | 87.10 | 66.70 | 81.30 | 85.50 | 84.40 | 74.00 | 77.91 |
| | | | | Training-free Methods | | | | |
| RIGID | 82.07 | 68.53 | 73.33 | **86.23** | 68.80 | 80.63 | **93.13** | 78.96 |
| ConV | **85.13** | 74.53 | **73.80** | 72.97 | 80.00 | **87.57** | 89.94 | **80.56** |

5,000 frames. With these images used as generated images and Laion serving as natural images, we further evaluate ConV's performance and compare it with baselines.

**Baselines and evaluation metrics.** We use training-free and training-based methods as baselines. For training-free methods, we take RIGID (He et al., 2024) and AEROBLADE (Ricker et al., 2024) as baselines. For training-based methods, we take DIRE (Wang et al., 2023a), CNNspot (Wang et al., 2020), Ojha (Ojha et al., 2023) and NPR (Tan et al., 2024) as baselines. For some baselines, we get the results reproted in their papers, including Frank (Frank et al., 2020), Durall (Durall et al., 2020), Patchfor (Chai et al., 2020), F3Net (Qian et al., 2020), SelfBland (Shiohara & Yamasaki, 2022), GANDetection (Mandelli et al., 2022), LGrad (Tan et al., 2023), ResNet-50 (He et al., 2016), DeiT-S (Touvron et al., 2021), Swin-T (Liu et al., 2021b), Spec (Zhang et al., 2019), LaRE[2] (Luo et al., 2024) and GramNet (Liu et al., 2020).

Following previous works, we mainly use the following metrics: (1) the average precision (AP) and (2) the area under the receiver operating characteristic curve (AUROC). Reproducing all baselines' results on some datasets with the same setting requires significant resources. Thus, we directly leverage the corresponding papers' results and report the classification accuracy (ACC).

**Implementation details.** In our experiments, we use the DINOv2 to instantiate $f_1(\cdot)$ and DINOv2's transformation[1] to realize $h(\cdot)$, as it is trained over a large-scale natural image dataset. There are four pre-trained DINOv2 models, i.e., ViT-S/14, ViT-B/14, ViT-L/14, and ViT-g/14, achieving exciting AUROC performance on ImageNet benchmark: 62.84, 78.58, 87.13, and 85.97, respectively. To balance detection performance and efficiency, we use DINOv2 ViT-L/14 in the following experiments. Meanwhile, We leverage data augmentations used in the training phase to realize the function $h(\cdot)$ in $f_2 = f_1 \circ h$, including geometric augmentation, color jitter, and Gaussian blur. Since data augmentation is randomized, to enhance performance, we can apply the function $n$ times to a single test image. As illustrated in Figure 4, increasing $n$ correlates with improved detection performance. However, to maintain detection efficiency, we set $n = 20$ in our experiments[2]. In practical applications, if multiple machines are available, we can leverage parallel processing to implement multiple

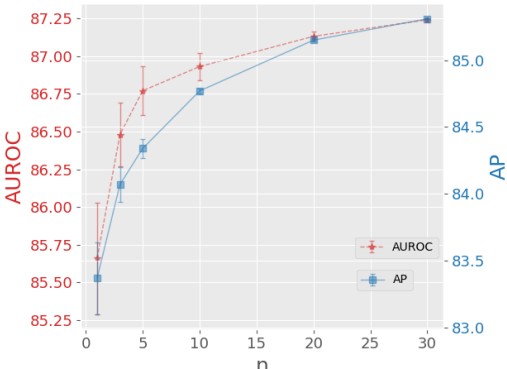

Figure 4: ConV with multiple forward passes.

transformations in a single forward pass to achieve better detection performance. In our experiments, we report the average results under five different random seeds.

---

[1]Details can be found in Appendix A.3

[2]For a fair comparison, we also set $n = 20$ for our baseline method, RIGID.

Table 4: AI-generated image detection performance on Sora.

| Models | Methods | | | | | | | | | | | |
|---|---|---|---|---|---|---|---|---|---|---|---|---|
| | CNNspot | | Ojha | | NPR | | AEROBLADA | | RIGID | | ConV | |
| | AUROC | AP | AUROC | AP | AUROC | AP | AUROC | AP | AUROC | AP | AUROC | AP |
| Sora | 52.85 | 53.29 | 77.06 | 80.69 | 51.92 | 50.25 | 57.13 | 58.00 | 84.22 | 81.98 | **87.74** | **88.85** |
| Open Sora | 50.14 | 51.38 | 67.05 | 68.67 | 50.25 | 51.84 | 55.79 | 62.37 | 73.12 | 75.56 | **82.84** | **85.24** |
| Average | 51.50 | 52.84 | 72.06 | 74.68 | 51.09 | 51.05 | 56.46 | 60.19 | 78.67 | 78.77 | **85.29** | **87.05** |

## 3.2 MAIN RESULT

**Comparison on public benchmarks.** We conduct comparative experiments across a comprehensive suite of standard benchmarks. As shown in Tables 1, 2, and 3, ConV achieves the best results under various scenarios, demonstrating its effectiveness and robustness. Note that ConV, as a train-free approach, outperforms the existing training-based methods that are typically trained using numerous natural and generated images. Moreover, on the large-scale benchmark GenImage, these training-based methods exhibit relatively poor generalization capability, while ConV can detect generated images effectively, illustrating the effectiveness of the generalization ability of the proposed method.

**Comparison on Sora.** We further evaluate ConV's performance on videos generated by unknown models. As shown in Table 4, ConV demonstrates the best performance on images generated by these unknown generative models, outperforming training-based methods. These results highlight the effectiveness and robustness of the proposed ConV.

**Illustration of the effectiveness.** We visualize the features of natural/real image $\mathbf{x}_n$ and generated/fake image $\mathbf{x}_g$ as well as the features of their augmented versions, i.e., $h(\mathbf{x}_r)$ and $h(\mathbf{x}_g)$. We extract features of $\mathbf{x}_n$, $\mathbf{x}_g$, $h(\mathbf{x}_n)$ and $h(\mathbf{x}_g)$ using DINOv2 and use t-SNE to visualize these features. To avoid the effect of class, all images are sampled from the same class for visualization. As shown in Figure 5, the conclusions are mainly twofold. First, the features of natural ($\mathbf{x}_n$) and augmented ($h(\mathbf{x}_n)$) images can be distinguished from those of generated images and their augmented versions, showing DINOv2's ability to differentiate between real and generated images. This provides a promising direction to leverage DINOv2 for

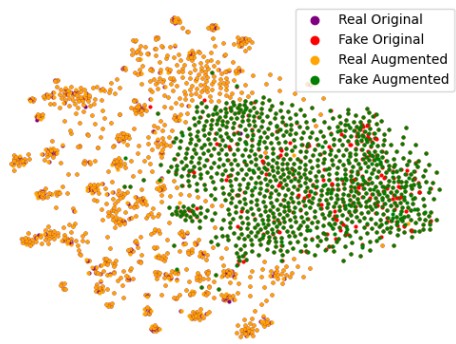

Figure 5: t-SNE visualization of features extracted by DINOv2.

AI-generated image detection. Second, the separation between a generated image and its augmented version in the representation space is more pronounced than that of real images. The feature of $h(\mathbf{x}_n)$ is similar to that of $\mathbf{x}_n$, i.e., features of $h(\mathbf{x}_n)$ substantially overlap with those of the natural image $\mathbf{x}_n$. In contrast, the features of $h(\mathbf{x}_g)$ generally fail to fully encompass those of the generated images $\mathbf{x}_g$. Aligning with this characteristic, ConV effectively distinguishes natural and generated images by calculating feature similarity between the original and augmented images. This is consistent with

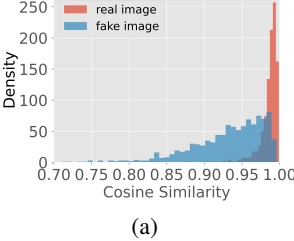

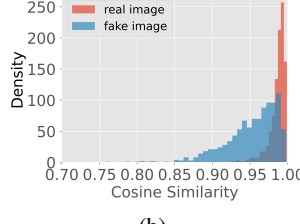

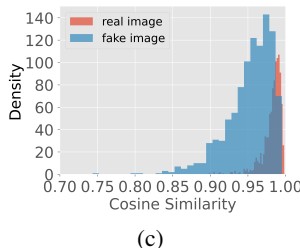

(a)                      (b)                      (c)

Figure 6: Cosine similarity between features of original image $\mathbf{x}$ and the transformed version $h(\mathbf{x})$, where fake images are generated by a) BigGAN, b) ADM, and c) DDPM.

the conclusion from Figure 6 showing the similarity between features of $\mathbf{x}$ and $h(\mathbf{x})$.

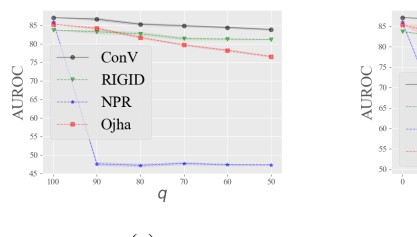 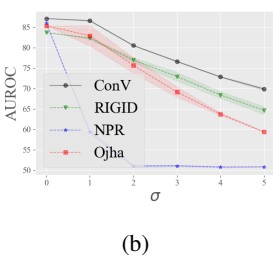 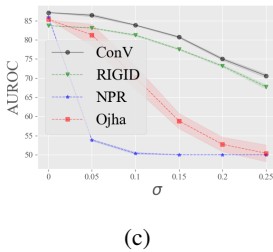

(a)        (b)        (c)

Figure 7: AI-generated image detection performance under various practical perturbations, i.e., (a): JPEG compression; (b): Gaussian blur; and (c) Gaussian noise.

### 3.3 DISCUSSION

When deploying a detector to identify generated images, it is crucial to consider practical environments or even a threat model. Specifically, images are often perturbed in practical scenarios, affecting detection performance. For instance, JPEG compression is a common mechanism due to the spread of images on the Internet. Moreover, AI-generated images may undergo post-processing to evade detection mechanisms. If a detection method is sensitive to some perturbations, the vulnerability would limit the applications in many practical scenarios. Thus, robustness to various perturbations is an essential metric in generated image detection. To verify the robustness of the proposed ConV, we process both natural and generated images by introducing some degradation mechanisms. Unless otherwise stated, experiments are conducted on the ImageNet dataset.

Following previous works (Ricker et al., 2024; He et al., 2024), we evaluate the robustness of detectors in three perturbations, including JPEG compression (with quality $q$), Gaussian blur, and Gaussian noise (both with standard deviation $\sigma$). The results are given in Figure 7. We can see that ConV achieves the best performance. We find that training-free methods usually show better robustness than training-based methods. In particular, although NPR achieves promising results on clean images, its performance degrades drastically when the perturbation increases level. This may stem from its reliance on the relationship between pixels. Namely, various small perturbations can change its features, causing its performance to degrade drastically. In contrast, ConV leverages the generalization ability of the pre-trained self-supervised model and is robust under various perturbations, which makes it suitable for a wider range of applications.

We verify the efficacy of the proposed method using more pre-trained models with results in Appendix A.7. The results demonstrate that our method can be applied for various pre-trained models.

### 4 RELATED WORKS

Our work focuses on AI-generated image detection. Thus, we first discuss the achievement of previous works. Then, we introduce some basic concepts of manifold learning used in our orthogonality principle. Our method is inherently related to self-supervised learning, which is also discussed.

**Generated images detection.** With the rapid advancements in generative models (Brock et al., 2019; Ho et al., 2020), the generation of highly realistic images has become increasingly feasible, thereby creating an urgent demand for effective algorithms to detect such generated images. Previous work (Frank et al., 2020; Marra et al., 2018) has usually focused on training a specialized binary classification neural network to distinguish between natural and generated images. CNNspt (Wang et al., 2020) finds that with specific data augmentation, a standard image classifier trained on Pro-GAN is able to generalize to other architectures. However, UniversalFakeDetect (Ojha et al., 2023) shows that the generalizability does not extend to unseen families of generative models. To this end, they propose to train classifiers in CLIP's (Radford et al., 2021) representation space to obtain stronger generalisability. DIRE (Wang et al., 2023a) uses the reconstruction error of an image on a diffusion model to train the classifier. NPR (Tan et al., 2024) leverages neighboring pixel relationships to elucidate the differences between natural and generated images. However, training-based approaches often suffer from generalizability issues and high computational costs. To address these limitations, several training-free methods have recently been proposed. AEROBLADE (Tan et al.,

2024) performs the detection by calculating the reconstruction error with the autoencoder used in latent diffusion models (Rombach et al., 2022). However, understanding the underlying mechanisms that enable these approaches to perform well on images generated by unknown generative models remains challenging. On the contrary, our method explicitly maps how the generated images are detected. Thus, exhibiting good generalization performance on images generated by unknown models is in line with expectations. Fortunately, our experiments on images generated by Sora and OpenSora provide effective support, see Table 4.

**Manifold learning.** Manifold learning Cayton et al. (2008) assumes that real-world data presented in high dimensional spaces are expected to concentrate in the vicinity of a manifold $\mathcal{M}$ of much lower dimensionality, embedded in high dimensional space. Namely, the probability mass tends to concentrate in regions with significantly lower dimensionality than the original space in which the data resides Bengio et al. (2013). In this context, tangent directions/spaces of the manifold. The tangent space of the manifold changes as the point-of-interest moves on the manifold, as shown in Figure 2. The local tangent space at a point on the manifold can be considered as capturing locally valid transformations, i.e., transformed points are still on the data manifold. Intuitively, a well-trained model is invariant to transformations along the tangent space Simard et al. (1991), which is mathematically equal to the orthogonality between vectors from the tangent space and the gradient of the model's loss with respect to the input, i.e., Eq. 8.

**Self-supervised learning.** Self-supervised learning Liu et al. (2021a) leverages input data as supervision, aiming to extract representations benefiting downstream tasks. In this regard, contrastive learning has become a dominant component in self-supervised learning. As a classical method (Becker & Hinton, 1992), contrastive learning aims to match the representations of the original and augmented images. Contrastive predictive coding (Oord et al., 2018) is one of the pioneering approaches to including contrastive learning in self-supervised learning. SimCLR (Chen et al., 2020) demonstrates the importance of large batches and negative pairs, while BYOL (Grill et al., 2020) removes the need for negative samples through self-distillation. SwAV (Caron et al., 2020) introduces a clustering-based approach, improving learning without explicit pairings. MoCo (He et al., 2020) enhances efficiency with a memory bank for negative sampling. DINO (Caron et al., 2021) further refined this by leveraging self-distillation and attention mechanisms, leading to stronger representations. DINOv2 (Oquab et al., 2024) pushed these advancements with large amount curated data from diverse sources. These methods have shown success in modeling the natural data distribution, especially for the robustness against data transformations. Thus, ConV exploits the property of performing contrastive learning on a large amount of natural data to realize the introduced functions. In this work, we mainly leverage DINOv2 (Oquab et al., 2024) as the introduced function $f_1(\cdot)$.

## 5 CONCLUSION AND LIMITATION

In this work, we propose ConV, a novel framework for detecting AI-generated images. Unlike existing methods that rely heavily on substantial datasets of natural and generated images, Conv relies solely on the natural image distribution. This is achieved by designing two functions whose outputs exhibit consistency for natural images but significant inconsistency for generated images. Extensive experiments on diverse benchmarks and images generated by a currently inaccessible model, i.e., Sora, have demonstrated ConV's superior performance.

**Limitation.** **1)** Although the proposed orthogonal principle provides an approach for designing various types of functions and its validity is widely supported by extensive empirical studies, we have not provided formal proof of the convergence of the generalization risk within the context of AI-generated image detection. Thus, our future work will focus on establishing the theoretical foundations of the generalization of our approach. **2)** Although we consider a threat model to verify the robustness of detectors, we have not provided an aggressive scenario where generative models are trained to minimize the inconsistency between $f_1$ and $f_2 = f_1 \circ h$. Thus, we will investigate the potential of integrating effective, robust, and efficient detection methods into the training process of generative models to make the generated images more realistic. **3)** Despite numerous empirical studies validating the effectiveness of the proposed CONV, the impact of scaling up the self-supervised model on the performance of detecting generated images remains to be explored since collecting a larger dataset and training an expanded self-supervised model are beyond the scope of this study. Moreover, future work is needed to explore how the performance of ConV will be affected if self-supervised models are trained on AI-generated images.

## ETHIC STATEMENT

This paper does not raise any ethical concerns. This study does not involve any human subjects, practices to data set releases, potentially harmful insights, methodologies and applications, potential conflicts of interest and sponsorship, discrimination/bias/fairness concerns, privacy and security issues, legal compliance, and research integrity issues.

## REPRODUCIBILITY STATEMENT

We summarize our efforts below to facilitate reproducible results:

- **Theoretical results.** A clear statement of the theoretical results can be found in Appendix A.1.
- **Datasets.** We use publicly available datasets, which are described in detail in Appendix A.4.
- **Open Source.** Code will be available once the paper is accepted.

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

## A APPENDIX

### A.1 DERIVATION FOR INCONSISTENCY

Here, we give the detailed derivation of Eq. 4. We expand these two functions at $\mathbf{x} := \mathbf{x}_{\mathcal{M}}(\mathbf{x}_g)$ for a given generated image $\mathbf{x}_g$,

$$f_1(\mathbf{x}_g) = f_1(\mathbf{x}) + \nabla f_1(\mathbf{x})^\top (\mathbf{x}_g - \mathbf{x}), \quad f_2(\mathbf{x}_g) = f_2(\mathbf{x}) + \nabla f_2(\mathbf{x})^\top (\mathbf{x}_g - \mathbf{x}), \quad (14)$$

where we neglect the higher-order approximation error.

The inconsistency between generated images can be formalized by,

$$\delta(\mathbf{x}_g) = |f_1(\mathbf{x}) - f_2(\mathbf{x}) + (\nabla f_1(\mathbf{x}) - \nabla f_2(\mathbf{x}))^\top (\mathbf{x}_g - \mathbf{x})| = |(\nabla f_1(\mathbf{x}) - \nabla f_2(\mathbf{x}))^\top (\mathbf{x}_g - \mathbf{x})|, \quad (15)$$

where the equation holds because of $\delta(\mathbf{x}) = f_1(\mathbf{x}) - f_2(\mathbf{x}) = 0$. Then, we have

$$\delta(\mathbf{x}_g) = |(\nabla f_1(\mathbf{x}) - \nabla f_2(\mathbf{x}))^\top (\mathbf{x}_g - \mathbf{x})| \geq ||\nabla f_1(\mathbf{x})^\top (\mathbf{x}_g - \mathbf{x})| - |\nabla f_2(\mathbf{x})^\top (\mathbf{x}_g - \mathbf{x})||. \quad (16)$$

### A.2 SOFTWARE AND HARDWARE

We use python 3.8.16 and Pytorch 1.12.1, and seveal NVIDIA GeForce RTX-3090 GPU and NVIDIA GeForce RTX-4090 GPU.

### A.3 DETAILS OF TRANSFORMATIONS

We follow the data augmentation strategy used when training DINOv2 with a combination of HorizontalFlip, ColorJitter, and GaussianBlur. For ColorJitter, brightness, contrast, saturation, and hue are randomly adjusted with a factor in the ranges of [0.88,1.12],[0.88,1.12],[0.94,1.06], and[0.97,1.03], respectively. For GaussianBlur, the kernel size is set to 9×9, and the variance is randomly selected in [0.7,1].

### A.4 DETAILS OF DATASETS

**IMAGENET.** The real images and generated images can be obtained at `https://github.com/layer6ai-labs/dgm-eval`. The images are provided by (Stein et al., 2023). The resolution of real images and generated images are $256 \times 256$. We crop the image randomly to $224 \times 224$ resolution. The generated images include:

- ADM, FID = 11.84.
- ADMG, FID = 5.58.
- BigGAN, FID = 7.94.
- DiT-XL-2, FID = 2.80.
- GigaGAN, FID=4.16.
- LDM, FID=4.29.
- StyleGAN-XL, FID=2.91.
- RQ-Transformer, FID=9.71.
- Mask-GIT, FID=5.63.

**LSUN-BEDROOM.** The real images and generated images can be obtained at `https://github.com/layer6ai-labs/dgm-eval`. The images are provided by (Stein et al., 2023). The resolution of real images and generated images are $256 \times 256$. We crop the image randomly to $224 \times 224$ resolution. The generated images include:

- ADM, FID=2.20.
- DDPM, FID=5.18.
- iDDPM, FID=4.54.

Table 5: AI-generated image detection performance on ImageNet.

| Methods | ADM | | ADMG | | LDM | | DiT | | BigGAN | | GigaGAN | | StyleGAN XL | | RQ-Transformer | | Mask GIT | | Average | |
|---|---|---|---|---|---|---|---|---|---|---|---|---|---|---|---|---|---|---|---|---|
| | AUROC | AP | AUROC | AP | AUROC | AP | AUROC | AP | AUROC | AP | AUROC | AP | AUROC | AP | AUROC | AP | AUROC | AP | AUROC (↑) | AP (↑) |
| Random rotation (-90-90 degrees) | 74.43 | 75.23 | 67.44 | 66.45 | 65.60 | 65.12 | 65.47 | 65.71 | 75.20 | 76.89 | 71.72 | 74.41 | 74.66 | 77.13 | 76.36 | 77.62 | 71.21 | 72.95 | 71.34 | 72.39 |
| Random rotation (-45-45 degrees) | 79.91 | 79.12 | 71.61 | 68.80 | 69.65 | 66.87 | 70.03 | 68.12 | 82.11 | 81.95 | 79.21 | 79.65 | 83.09 | 83.58 | 82.79 | 82.03 | 77.91 | 77.54 | 77.37 | 76.41 |

- StyleGAN, FID=2.65.

- Diffusion-Projected GAN, FID=1.79.

- Projected GAN, FID=2.23.

- Unleashing Transformers, FID=3.58.

**GenImag** The real images and generated images can be obtained at `https://github.com/GenImage-Dataset/GenImage`. The images are provided by (Zhu et al., 2023). The real images come from ImageNet, and different images have different resolutions. Following (Stein et al., 2023),we resize the image to $256 \times 256$ resolution and adjust its format to keep the same with the generated images, then we randomly crop it to $224 \times 224$ resolution to extract features. The generated images include:

- Midjourney. The resolution of images generated by Midjourney is $1024 \times 1024$, and we randomly crop them to $224 \times 224$ resolution.

- SD V1.4. The resolution of images generated by SD V1.4 is $512 \times 512$, and we randomly crop them to $224 \times 224$ resolution.

- SD V1.5. The resolution of images generated by SD V1.5 is $512 \times 512$, and we randomly crop them to $224 \times 224$ resolution.

- ADM. The resolution of images generated by SD V1.5 is $256 \times 256$, and we randomly crop them to $224 \times 224$ resolution.

- GLIDE. The resolution of images generated by SD V1.5 is $256 \times 256$, and we randomly crop them to $224 \times 224$ resolution.

- Wukong. The resolution of images generated by SD V1.5 is $512 \times 512$, and we randomly crop them to $224 \times 224$ resolution.

- VQDM. The resolution of images generated by SD V1.5 is $256 \times 256$, and we randomly crop them to $224 \times 224$ resolution.

- BigGAN. The resolution of images generated by SD V1.5 is $128 \times 128$, and we fill them with zero pixels to $224 \times 224$ resolution.

Note that, in the original GenImage dataset, the natural images are all saved in *jpg* format, while the generated images are all saved in *png* format, and this unwanted bias will result in unrealistic detection performance. This is also discussed in AEROBLADE (Ricker et al., 2024). Therefore, we follow (Stein et al., 2023) to convert all the natural images to *png* format and pre-scaled the images to $256 \times 256$. Since the generated images are already in *png* format, we don't do anything with them beforehand.

## A.5 RESULTS OF USING OTHER DATA TRANSFORMATIONS.

In our experiments, we leverage data augmentations used in the training phase, including geometric augmentations, color jitter, and Gaussian blur. We further conduct comparison experiments using data augmentations which is not used during training, such as random rotation. The experiments are conducted on the ImageNet benchmark. As shown Table 5, using data transformations not seen during training does not result in good detection performance. Since the rotations were not used for data augmentation during training, using them to perform ConV during testing could not achieve good detection performance.

## A.6 RESULTS ON CLIP

In our paper, we use DINOv2 for all of our experiments. We further use CLIP for comparison experiments. We note that the authors only used randomly crop as data augmentation when training

Table 6: AI-generated image detection performance on ImageNet.

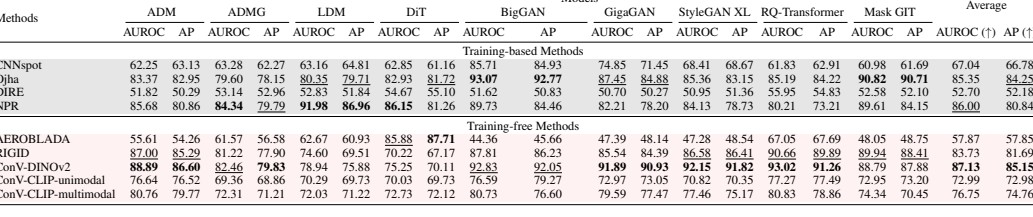

| Methods | ADM AUROC | AP | ADMG AUROC | AP | LDM AUROC | AP | DiT AUROC | AP | BigGAN AUROC | AP | GigaGAN AUROC | AP | StyleGAN XL AUROC | AP | RQ-Transformer AUROC | AP | Mask GIT AUROC | AP | Average AUROC (↑) | AP (↑) |
|---|---|---|---|---|---|---|---|---|---|---|---|---|---|---|---|---|---|---|---|---|
| *Training-based Methods* | | | | | | | | | | | | | | | | | | | | |
| CNNspot | 62.25 | 63.13 | 63.28 | 62.27 | 63.16 | 64.81 | 62.85 | 61.16 | 85.71 | 84.93 | 74.85 | 71.45 | 68.41 | 68.67 | 61.83 | 62.91 | 60.98 | 61.69 | 67.04 | 66.78 |
| Ojha | 83.37 | 82.95 | 79.60 | 78.15 | 80.35 | 79.71 | 82.93 | 81.72 | 93.07 | 92.77 | 87.45 | 84.88 | 85.36 | 83.15 | 85.19 | 84.22 | 90.82 | 90.71 | 85.35 | 84.25 |
| DIRE | 51.82 | 50.29 | 53.14 | 52.96 | 52.83 | 51.84 | 54.67 | 55.10 | 51.62 | 50.83 | 50.70 | 50.27 | 50.95 | 51.36 | 55.95 | 54.83 | 52.58 | 52.10 | 52.70 | 52.18 |
| NPR | 85.68 | 80.86 | 84.34 | 79.79 | 91.98 | 86.96 | 86.15 | 81.26 | 89.73 | 84.46 | 82.21 | 78.20 | 84.13 | 78.73 | 80.21 | 73.21 | 89.61 | 84.15 | 86.00 | 80.84 |
| *Training-free Methods* | | | | | | | | | | | | | | | | | | | | |
| AEROBLADA | 55.61 | 54.26 | 61.57 | 56.58 | 62.67 | 60.93 | 85.88 | 87.71 | 44.36 | 45.66 | 47.39 | 48.14 | 47.28 | 48.54 | 67.05 | 67.69 | 48.05 | 48.75 | 57.87 | 57.85 |
| RIGID | 87.00 | 85.29 | 81.22 | 77.90 | 74.60 | 69.51 | 70.22 | 67.17 | 87.81 | 86.23 | 85.54 | 84.39 | 86.58 | 86.41 | 90.66 | 89.89 | 89.94 | 88.41 | 83.73 | 81.69 |
| ConV-DINOv2 | 88.89 | 86.60 | 82.46 | 79.83 | 78.94 | 75.88 | 75.25 | 70.11 | 92.83 | 92.05 | 91.89 | 90.93 | 92.15 | 91.82 | 93.02 | 91.26 | 88.79 | 87.88 | 87.13 | 85.15 |
| ConV-CLIP-unimodal | 76.64 | 76.52 | 69.36 | 68.86 | 70.29 | 69.73 | 70.03 | 69.73 | 76.59 | 79.27 | 72.97 | 73.05 | 70.82 | 70.35 | 77.27 | 77.49 | 72.95 | 73.20 | 72.99 | 72.98 |
| ConV-CLIP-multimodal | 80.76 | 79.77 | 72.31 | 71.21 | 72.03 | 71.22 | 72.73 | 72.12 | 80.73 | 76.60 | 79.59 | 77.47 | 77.46 | 75.17 | 80.83 | 78.86 | 74.34 | 70.45 | 76.75 | 74.76 |

Table 7: AI-generated image detection performance with different pre-trained models.

| Methods | MoCo AUROC | AP | SwAV AUROC | AP | DINO AUROC | AP | CLIP AUROC | AP | DINOv2 AUROC | AP |
|---|---|---|---|---|---|---|---|---|---|---|
| RIGID | 58.67 | 57.05 | 61.88 | 61.41 | 67.98 | 66.14 | 66.50 | 65.32 | 83.73 | 81.69 |
| ConV | **68.43** | **67.65** | **74.71** | **73.48** | **71.91** | **69.46** | **72.99** | **72.98** | **87.13** | **85.15** |

CLIP. Therefore, when implementing ConV with CLIP, we also only use random crop. As shown in Table 6, using CLIP to implement ConV does not achieve good performance. We speculate that this difference comes from the training methodology. CLIP learns features using image captions as supervision, which may make the features more focused on semantic information, whereas DINOv2 learns features only from images, which makes it more focused on the images themselves, and thus better able to capture the subtle differences between the real image and the generated image. In addition to this, the fact that CLIP only uses random crop as data augmentation may also contribute to the poor performance of ConV.

The results show that our method performs relatively worse when using CLIP. To overcome this limitation, we revisit our methodology, i.e., verifying the consistency between outputs of two functions.

As shown in Eq. (13), we derive the cosine similarity metric between image features from a self-supervised learning objective function. However, CLIP employs a different objective function, namely, calculating the similarity between text and image features. Thus, the proposed cosine similarity between image features may not be a good realization of these two functions' output, limiting the generalization capability for generated image detection. We conjecture the difference between the projection of the visual features of the original image and the visual features of the transformed image on their corresponding text features would be a good metric. The reason is as follows: The function to calculate the similarity between text and image features can be regarded as a function. Thus, we should calculate the difference in inter-modality similarity rather than the similarity between original and transformed images. To verify the point, we conduct experiments using the corrected realization of $f_1$ and $f_2$, i.e., a corrected metric to verify the consistency. The results below show that the modified approach outperforms the original metric.

These results show that using the modified metric for detection greatly improves the performance of CLIP-based methods model, achieving performance comparable with Dinov2-based methods. Hence, we believe our work provides a novel approach to calculating the difference between two functions without focusing on the differences in similarities between two image features.

### A.7 RESULT ON MORE PRE-TRAINED MODELS

Besides CLIP, we conduct experiments using the MoCo (He et al., 2020), SwAV (Caron et al., 2020), and DINO (Caron et al., 2021). The results are reported in Table 7. These results show that our method can be applied to various backbones.

