# OpenReview forum: "Consistency Verification for Detecting AI-Generated Images"
_ICLR.cc/2025/Conference — Submitted to ICLR 2025_

### Official Review · Reviewer_qU6e · 2024-10-18

**Soundness:** 2
**Presentation:** 1
**Contribution:** 1
**Rating:** 3
**Confidence:** 5

**Summary:**

This paper proposes a consistency verification approach that does not require AI-generated images for training. The key component of the design is two functions: the representation function f1 based on the self-supervised pre-trained model (e.g., Dinov2). The transformation function h is also based on the self-supervised model's data augmentation. The author hypothesizes that the deviation after the transformation function in the representation space will be smaller in the real image, and larger in the AI-generated images.

**Strengths:**

1) I like the setting where the pre-trained foundation model is used for the prediction instead of exhausting training on AI-generated or real data.
2) Experiment on SORA benchmark is also interesting.

**Weaknesses:**

1) I feel the novelty of the paper is weak. The paper borrows lots of ideas from the RIGID, the baseline paper. RIGID measures cosine similarity between the dinov2 representation of the original image and the dinov2 representation of the deviated image through Gaussian Noise Augmentation. This paper does the same except changing cosine similarity to l2 distance, and Gaussian augmentation to training augmentations (e.g. Gaussian Blur, Colorjitter). Hence, I feel like this paper is just an ablation study of the RIGID paper, and the contributions should go to the original paper (e.g. applying Dinov2, and proposing a similar consistency function).

2) Experiment results also feel weak. The proposed method does not outperform RIGID in all cases. Aligned to 1), I feel like this paper should at least outperform RIGID in all experiments, not only on the average but also in all AI-generated data. Maybe more deeper analysis will be helpful.

3) Section 2 is hard to read. 2.1~2.3 seems to overexplain general ideas of manifold hypothesis, which is not new. In my opinion, this is just a longer extraction of "This model is trained on the real data so it will show robust output on the real data when the training data augmentation is applied". The final consistency term form has been applied to various binary classification tasks ([1],[2]). Maybe a deeper explanation of why this explanation is necessary will be helpful.

4) The experiment results of RIGID are not the same, but deteriorated from the original paper, especially in the ImageNet dataset. I wonder why such a discrepancy occurs, given that the performance gap between the proposed method and RIGID is not so large. This is more puzzling since the performance of RIGID reported in this paper should be improved by using larger $n=20$. I hope the elaboration on this result will be beneficial to resolve my concerns.

Overall, I find the paper's novelty weak since RIGID is also based on the distance between the original dinov2 representation and the dinov2 representation of the perturbed data, given that RIGID is the main baseline of this paper.

[1]  Likelihood Ratios for Out-of-Distribution Detection, NIPS 2019

[2] Likelihood Regret: An Out-of-Distribution Detection Score For Variational Auto-encoder, NeurIPS 2020

**Questions:**

See weakness for the main issues.

1) I did not quite understand of the data processing procedure of the GenImage dataset on Page 15. I feel like the authors cropped the AI-generated images in 224 by 224 dimensions, but treated ImageNet data differently by resizing first, and then cropping it to 224 by 224 size.
Did I understand it correctly? If it is, why the processing method is different in AI-generated and real images?

2) I am curious how AEROBLADE will perform in the GenImage dataset. (The table has typos by the way)

3) While this is minor, I would like to see the runtime analysis of the proposed method, given that the paper utilizes lots of ensembles.

---

> ### Author Response · Authors · 2024-11-23
>
> We would like to thank Reviewer qU6e for taking the time to review our work. Please see our detailed responses to your constructive comments below.
>
>
> > Weakness 1: I feel the novelty of the paper is weak. The paper borrows lots of ideas from the RIGID. RIGID measures cosine similarity between the DINOv2 representation of the original image and the DINOv2 representation of the deviated image through Gaussian Noise Augmentation. This paper does the same except changing cosine similarity to l2 distance, and Gaussian augmentation to training augmentations (e.g. Gaussian Blur, Colorjitter). Hence, I feel like this paper is just an ablation study of the RIGID paper, and the contributions should go to the original paper (e.g. applying DINOv2, and proposing a similar consistency function).
>
> Response:
>
> We respect your understanding of our method through your perspective. Please let us list points to highlight the differences from the mentioned RIGID.
>
> First, our method is based on the derived orthogonal principle, which ensures the generalization performance of our method. In contrast, RIGID is based on the empirical observations, leading potential risk of RIGID. Namely, the performance of RIGID could drastically degrade when generated images are perturbed with Gaussian noise. Detailed results and discussions can be found in the responses to obBE’s Weakness 2.
>
> Second, the realization of our orthogonal principal is not merely focusing the similarity between images and their transformed versions, as shown in the responses to Reviewer t4Ze [Question 1]. In contrast, as mentioned in your comments, RIGID mainly focuses on the similarity between features of images and images with Gaussian noise.
>
> Third, as mentioned in your comments, we compared our method with RIGID. This is because we appreciate the contribution of RIGID due to its simple yet effective training-free framework. We believe it would be a standard baseline in our community, so we set it as our baseline method.
>
> > Weakness 2:  Experiment results also feel weak. The proposed method does not outperform RIGID in all cases. Aligned to 1), I feel like this paper should at least outperform RIGID in all experiments, not only on the average but also in all AI-generated data. Maybe more deeper analysis will be helpful.
>
> Response:
>
> Due to the diversity of generative models, the various generated images may have different properties, and it is challenging for a single method to outperform recent SOTA work in all cases. Nevertheless, ConV outperforms RIGID on average across various benchmarks.
>
> > Weakness 3: Section 2 is hard to read. 2.1~2.3 seems to overexplain general ideas of manifold hypothesis, which is not new. In my opinion, this is just a longer extraction of "This model is trained on the real data so it will show robust output on the real data when the training data augmentation is applied". The final consistency term form has been applied to various binary classification tasks ([ref2],[ref3]). Maybe a deeper explanation of why this explanation is necessary will be helpful.
>
> Response:
>
> We apologize for any confusion. Accordingly, we would like to highlight the main points in Section 2.
>
> In Sec. 2.1, we give the general ideas of manifold hypothesis. In Sec. 2.2, we give the orthogonal principal, namely, the introduced two functions can detect generated images through the orthogonal principal: gradients of these two functions should lie within two mutually orthogonal spaces. In Sec. 2.3, we give a realization of the orthogonal principal based on the manifold hypothesis discussed in Sec. 2.1.
>
> We agree that OOD detection provides many outstanding methodologies, such as the consistency perspective [ref2, ref3]. However, researchers can able to draw inspirations from OOD detection. For instance, post-hoc OOD detection methods usually leverage pretrained models to detect OOD data, which will not be the reason to weaken the contribution of RIGID, namely, RIGID is an outstanding post-hoc approach to detect generated images using a pretrained model.

---

> ### Author Response · Authors · 2024-11-23
>
> > Weakness 4: The experiment results of RIGID are not the same, but deteriorated from the original paper, especially in the ImageNet dataset. I wonder why such a discrepancy occurs, given that the performance gap between the proposed method and RIGID is not so large. This is more puzzling since the performance of RIGID reported in this paper should be improved by using larger $n = 20$. I hope the elaboration on this result will be beneficial to resolve my concerns.
>
> Response:
>
> The authors of RIGID have not yet released their code, so we were unable to obtain information in their experimental section about the way their data is preprocessed and the strength of the Gaussian noise they added. Therefore, we used the same data preprocessing for their method as ours and set the Gaussian noise strength to 0.1 based on the results of the ablation experiments in their paper. These differences may have contributed to the performance gap.
>
> > Weakness 5: I did not quite understand of the data processing procedure of the GenImage dataset on Page 15. I feel like the authors cropped the AI-generated images in 224 by 224 dimensions, but treated ImageNet data differently by resizing first, and then cropping it to 224 by 224 size. Did I understand it correctly? If it is, why the processing method is different in AI-generated and real images?
>
> Response:
>
> Thanks for your comments. Your understanding is correct. In the original GenImage dataset, the natural images are all saved in *jpg* format, while the generated images are all saved in *png* format, and this unwanted bias will result in unrealistic detection performance, which is discussed in AEROBLADE [ref2]. Therefore, we follow [ref3] to convert all the natural images to *png* format and pre-scaled the images to 256*256. Since the generated images are already in *png* format, we don't do anything with them beforehand. We will include these descriptions in our revision.
>
> > Weakness 6: I am curious how AEROBLADE will perform in the GenImage dataset. (The table has typos by the way)
>
> Response:
>
> Thanks for the heads up, we've corrected it in the revised version. For AEROBLADE, since GenImage is too large, we don't have enough GPUs to run AEROBLADE on the full dataset at this moment. Therefore, for each class of generative models, we randomly pick 10,000 natural images and 10,000 generated images to test its performance. The results are as follows:
>
> |           | Midjourney | SD V1.5 | ADM   | GLIDE | Wukong | VQDM  | BigGAN | AVG   |
> |-----------|------------|---------|-------|-------|--------|-------|--------|-------|
> |           | ACC        | ACC     | ACC   | ACC   | ACC    | ACC   | ACC    | ACC   |
> | Rigid     | 82.07      | 68.53   | 73.33 | 86.23 | 68.80  | 80.63 | 93.13  | 78.96 |
> | AEROBLADE | 68.98      | 64.90   | 57.40 | 68.38 | 67.50  | 57.83 | 73.73  | 65.53 |
> | ConV      | 85.13      | 74.53   | 73.80 | 72.97 | 80.00  | 87.57 | 89.94  | 80.56 |
>
> > Weakness 7: While this is minor, I would like to see the runtime analysis of the proposed method, given that the paper utilizes lots of ensembles.
>
> Response:
>
> Thanks for your kind suggestion. As shown in the table below, we count the time used to detect 100 images. When using 20 ensembles, ConV takes longer detection time than AEROBLADE. Of course, this issue can be addressed by parallelization or reducing the number of ensembles. When reducing the number of ensembles, the performance suffers a slight degradation, as shown in Figure 4.
>
> |           | Time |
> |-----------|------|
> | AEROBLADE | 17s  |
> | ConV      | 26s  |
>
> Reference:
>
> [ref1] RIGID: A Training-Free and Model-Agnostic Framework for Robust AI-Generated Image Detection. arxiv, 2024.
>
> [ref2]: AEROBLADE: Training-Free Detection of Latent Diffusion Images Using Autoencoder Reconstruction Error. CVPR, 2024.
>
> [ref3]: Exposing flaws of generative model evaluation metrics and their unfair treatment of diffusion models. NeurIPS, 2024.

---

> > ### Comment · Reviewer_qU6e · 2024-11-24
> > **Response to the rebuttal.**
> >
> > Thank you for your rebuttal.
> >
> > I do not feel my main concerns about novelty have been resolved, given that the authors only responded by rephrasing the claims of the submitted paper (W1, W3).
> >
> > While the rebuttal also remains further questions (e.g. why AEROBLADE scores low accuracy in even detecting SDv1.5 generated datasets, RIGID stated their intensity parameter as 0.05 in the Appendix as opposed to the author's response of W4 ), I feel like the paper's main contribution is superficial.
> >
> > I stick to my current rating.

---

> > > ### Author Response · Authors · 2024-11-24
> > >
> > > Thank you for your feedback.
> > >
> > > >Q1: I do not feel my main concerns about novelty have been resolved, given that the authors only responded by rephrasing the claims of the submitted paper (W1, W3).
> > >
> > > A1: We would appreciate it if you could list specific reasons why the listed contributions as not valid. We will incorporate your specific and detailed feedback to improve the quality of our work.
> > >
> > > * Our method is based on the derived orthogonal principle: **the gradients of two functions need to lie within two mutually orthogonal subspaces for generated image detection**. This ensures the generalization performance of our method. In contrast, RIGID is based on the empirical observations, leading potential risk of RIGID. Namely, **the performance of RIGID could drastically degrade when generated images are perturbed with Gaussian noise**. Detailed results and discussions can be found in the responses to obBE’s Weakness 2. Besides, the empirical observation of RIGID may not be applicable to all cases. As shown in the table below, we replace all the natural images in the ImageNet benchmark with the images from LAION [ref1]. In this case, RIGID is unable to distinguish between natural and generated images. On the contrary, ConV maintains the discriminative ability.
> > >
> > > * The realization of our orthogonal principal is not merely focusing the similarity between images and their transformed versions, as shown in the responses to Reviewer t4Ze [Question 1]. In contrast, as mentioned in your comments, RIGID mainly focuses on the similarity between features of images and images with Gaussian noise.
> > >
> > > |       | AUROC | AP    |
> > > |-------|-------|-------|
> > > | RIGID | 55.80 | 54.48 |
> > > | ConV  | 71.54 | 65.78 |
> > >
> > > >Q2: Why AEROBLADE scores low accuracy in even detecting SDv1.5 generated datasets?
> > >
> > > A2: The resolution of the images used in AEROBLADE is larger than 512 * 512 in order to crop the images to 512 * 512 resolution. However, in GenImage, this obviously can't be achieved:
> > > The natural images come from ImageNet. Their resolution is not regular, and there are many images that do not satisfy the requirement. Among the generated images, only Midjourney, SD v1.5, and Wukong satisfy the requirement of resolution larger than 512 * 512.
> > >
> > > In our experiments, we scale or crop the image to a resolution of 224 * 224 before detection. We guess this preprocessing operation is the reason for AEROBLADE's poor performance. This may result from its sensitivity to image preprocessing operations. Note that, in RIGID's paper, they report similar results to ours.
> > >
> > > >Q2: RIGID stated their intensity parameter as 0.05 in the Appendix as opposed to the author's response of W4.
> > >
> > > A2: We run RIGID on ImageNet with noise intensity as 0.05. The results are shown in the table below. We can see that RIGID is relatively robust to the intensity of Gaussian noise. This is consistent with the results of ablation experiments in RIGID.
> > >
> > > |                                       | ADM   |       | ADMG  |       | Big GAN |       | DiT   |       | Giga GAN |       | LDM   |       | Mask-GIT |       | RQ-Transformer |       | StyleGAN-XL |       | Avg   |       |
> > > |---------------------------------------|-------|-------|-------|-------|---------|-------|-------|-------|----------|-------|-------|-------|----------|-------|----------------|-------|-------------|-------|-------|-------|
> > > |                                       | AUROC | AP    | AUROC | AP    | AUROC   | AP    | AUROC | AP    | AUROC    | AP    | AUROC | AP    | AUROC    | AP    |                |       |             |       | AUROC | AP    |
> > > | ConV               | 88.89 | 86.60 | 82.46 | 79.83 | 92.83   | 92.05 | 75.25 | 70.11 | 91.89    | 90.93 | 78.94 | 75.88 | 88.79    | 87.88 | 93.02          | 91.26 | 92.15       | 91.82 | 87.13 | 85.15 |
> > > | RIGID ($\lambda = 0.1$)  |87.00 | 85.29 | 81.22 | 77.90 | 87.81   | 86.23 | 70.22 | 67.17 | 85.54    | 84.39 | 74.60 | 69.51 | 89.94    | 88.41 | 90.66         | 89.89 | 89.94       | 88.41 | 83.73 | 81.69 |
> > > | RIGID ($\lambda = 0.05$) | 86.05 | 83.22 | 80.98 | 77.18 |  88.06  | 85.95 | 71.56 | 67.96 |84.72    | 82.23 | 74.44 | 70.22 | 90.43    | 89.11 | 89.43          | 88.22 |  84，82      | 83.12 | 83.39 | 80.80 |
> > >
> > > Reference:
> > >
> > > [ref1]: LAION-400M: Open Dataset of CLIP-Filtered 400 Million Image-Text Pairs.

---

> > > ### Author Response · Authors · 2024-11-27
> > >
> > > Dear Reviewer #qU6e,
> > >
> > > Thank you very much for your time and valuable feedback.
> > >
> > > Here is a summary of our response for your convenience:
> > >
> > > **(1) Novelty Issue:** The main difference between ConV and RIGID is that we focus on designing a pair of functions to verify consistency, rather than just feature similarity between images and their transformed versions. Our theory suggests that the gradients of the two designed functions should lie within two mutually orthogonal subspaces for generated image detection. Based on this principle, we can design various verification functions for different models. This consistency verification-based approach offers good robustness and generalizability.
> > >
> > > **(2) Issues with AEROBLADE's performance:** The poor performance of AEROBLADE is due to the resolution difference between the images in the GenImage dataset and those used in AEROBLADE's paper. This discrepancy leads to different data preprocessing steps.
> > >
> > > **(3) Issues with RIGID's hyperparameter:** We re-ran the experiment with a noise intensity of 0.05. However, the experimental results did not show significant improvement.
> > >
> > > We understand that you're busy, but as the window for responding and revising the paper is closing, would you mind reviewing our response and confirming if you have any further questions? We would be happy to provide additional answers or revisions if needed.
> > >
> > > Best regards and thanks,
> > > Authors of #8640

---

### Official Review · Reviewer_obBE · 2024-11-04

**Soundness:** 2
**Presentation:** 2
**Contribution:** 1
**Rating:** 3
**Confidence:** 4

**Summary:**

This paper presents a AI-generated images detection method called Consistency Verification (ConV). Unlike traditional approaches that rely on binary classifiers trained on both natural and AI-generated images, ConV is training-free. The key innovation lies in introducing two functions that remain consistent for natural images but show inconsistency for AI-generated images.

**Strengths:**

Training-Free**: It eliminates the need for large datasets of generated images and the computational cost associated with training a classifier.
Theoretical support: The method's validity is explained the orthogonality principle to some extend.**

**Weaknesses:**

See questions.

**Questions:**

1. *Limited Theoretical Justification:** While the orthogonality principle provides some intuition, a more rigorous theoretical analysis of why and how the method works would strengthen the paper. For instance, There is no explanation for why the augmentations during DinoV2 training are orthogonal to its standard output? Moreover, the theoretical analysis in the paper does not provide any guidance for the choice of augmentations.
2. **Lack of Novelty:** The method proposed in this paper is based on Baseline RIGID, but it only replaces Gaussian noise with training enhancements, without much insightful design. In addition, it does not explain why these enhancements are better than Gaussian noise.
3. **Limited Implementation Details:** The paper does not give specific details of the data augmentation used, including the parameters of the data augmentation used, and whether a single augmentation or a combination of multiple augmentations is used.
4. **Single Backbone Model:** The paper only tested on DinoV2 and not on more (self-supervised) models. Does this mean that the scope of application of this method is limited?

---

> ### Author Response · Authors · 2024-11-23
>
> We thank Reviewer obBE for taking the time to review our work. Please see our detailed responses to your comments below.
>
> > Weakness 1:
> There is no explanation for why the augmentations during DinoV2 training are orthogonal to its standard output? Moreover, the theoretical analysis in the paper does not provide any guidance for the choice of augmentations.
>
> Response:
>
> We apologize for the misunderstanding. Our focus on orthogonality is mainly about the gradients of the introduced two functions $f_1$ and $f_2$. Namely, the gradient of $f_1$ w.r.t the input is orthogonal to the local tangent space, while the gradient of $f_2$ w.r.t the input lies in the local tangent space (c.f. Eqs. (8) and (11)). The reason of why the gradient of $f_2$ lies in the local tangent space is that we realize the function $h$ of $f_2$ using the transformation along local data manifold.
> Our theoretical analysis shows that we can realize the function $h$ using transformation along local data manifold. One simple yet effect approach is to employ data augmentations used during training phase. This is because data augmentations are one approach to move data point along local tangent space [ref1].
>
> > Weakness 2:
> The method proposed in this paper is based on Baseline RIGID, but it only replaces Gaussian noise with training enhancements, without much insightful design. In addition, it does not explain why these enhancements are better than Gaussian noise.
>
> Response:
>
> We apologize for the misunderstanding.
>
> First, **our theoretical analysis shows that the gradients of the introduced two functions need to lie within two mutually orthogonal subspaces**. Thus, we propose to leverage the transformation along its data manifold to realize the function $h$. It is known that data augmentation can be regarded as transformation along data manifold. Thus, we propose to use data augmentations to realize the function $h$. In contrast, **RIGID’s theoretical analysis shows that RIGID mainly focuses on the difference in gradient norm between natural and generated images for detection**.
>
> Second, our method is not based on [ref2]. As shown in the response to Reviewer t4Ze [Question 1], replacing the cosine similarity between images with consistency of projections of original image features and transformed image features on text features can significantly improve the performance on CLIP. This highlights the novelty and contribution of the proposed consistency verification framework. Namely, **our method does not focus on the difference in similarities between an image and its transformed version**. In contrast, **our work focuses on verifying the inconsistency between two functions that have gradients mutually orthogonal to each other**.
>
> Third, our method significantly outperforms the mentioned RIGID, especially on images generated by the inaccessible generative model Sora. This is based on our orthogonal principal. In contrast, RIGID is based on the observed sensitivity of the generated images to noise, leading to potential risks. For instance, users can evade detection by adding noise to the generated image. As shown in the table below, when noise is added to a generated image, RIGID can easily be tricked into identifying it as a natural image.
>
> |                                   | AUROC | AP    |
> |-----------------------------------|-------|-------|
> | RIGID on clean images                            | 83.73 | 81.69 |
> | ConV on clean images                            | 87.13 | 85.15 |
> | RIGID on  noisy generated images | *18.69* | *34.51* |
> | ConV on noisy generated images  | 91.25 | 90.53 |
>
> Fourth, we would like to highlight that the consistency verification is the insightful design of our work. Specifically, we can introduce various metrics and functions to calculate the inconsistency. For instance, designing metric for specific models, as shown in our response to Reviewer t4Ze [Question 1]. In addition, we can introduce various approaches to construct different these two functions. All these directions are based on our orthogonal principal, which is not given in the mentioned work RIGID.

---

> ### Author Response · Authors · 2024-11-23
>
> > Weakness 3: The paper does not give specific details of the data augmentation used, including the parameters of the data augmentation used, and whether a single augmentation or a combination of multiple augmentations is used.
>
> Response:
>
> Thanks for your comments. We follow the data augmentation strategy used when training DINOv2 with a combination of HorizontalFlip, ColorJitter, and GaussianBlur. For ColorJitter, brightness, contrast, saturation, and hue are randomly adjusted with a factor in the ranges of [0.88,1.12],[0.88,1.12],[0.94,1.06], and[0.97,1.03], respectively. For GaussianBlur, the kernel size is set to 9*9, and the variance is randomly selected in [0.7,1]. We'll add this detailed description to the revision.
>
> > Weakness 4:
> The paper only tested on DinoV2 and not on more (self-supervised) models. Does this mean that the scope of application of this method is limited?
>
> Response:
>
> We apologize for the misunderstanding. TWO models, DINOv2 (self-supervised model) and CLIP (multimodal model), are used to evaluate the effectiveness of our method.
> Inspired by your valuable questions, we conduct experiments using the MoCo, SwAV and DINO. The results are as follows. These results show that our method can be applied to various backbones. We will add these results to the revision.
>
> |       | MoCo  |       | SwAV  |       | DINO  |       | CLIP  |       | DINOv2 |       |
> |-------|-------|-------|-------|-------|-------|-------|-------|-------|--------|-------|
> |       | AUROC | AP    | AUROC | AP    | AUROC | AP    | AUROC | AP    | AUROC  | AP    |
> | Rigid | 58.67 | 57.05 | 61.88 | 61.41 | 67.98 | 66.14 | 66.50 | 65.32 | 83.73  | 81.69 |
> | ConV  | 68.43 | 67.65 | 74.71 | 73.48 | 71.91 | 69.46 | 72.99 | 72.98 | 87.13  | 85.15 |
>
> Reference:
>
> [ref1] Tangent prop-a formalism for specifying selected invariances in an adaptive network. NIPS, 1991.
>
> [ref2] RIGID: A Training-Free and Model-Agnostic Framework for Robust AI-Generated Image Detection. arxiv, 30 May 2024.

---

> ### Author Response · Authors · 2024-11-27
>
> Dear Reviewer #obBE,
>
> Thank you very much for your time and valuable feedback.
>
> Here is a summary of our response for your convenience:
>
> **(1) Issues regarding theoretical justification:** Our focus of orthogonality is mainly about the gradients of the introduced two functions $f_1$ and $f_2$. Namely, the gradient of $f_1$w.r.t the input is orthogonal to the local tangent space while the gradient of $f_2$ w.r.t the input lies in the local tangent space (c.f. Eqs. (8) and (11)). The reason of why the gradient of $f_2$ lies in the local tangent space is that we realize the function $h$ of $f_2$ using the transformation along local data manifold.
>
> Our theoretical analysis shows that we can realize the function $h$ using transformation along local data manifold. One simple yet effect approach is to employ data augmentations used during training phase. This is because data augmentations is one approach to move data point along local tangent space.
>
> **(2) Novelty Issue:** The main difference between ConV and RIGID is that we focus on designing a pair of functions to verify consistency, where the natural image exhibits high consistency on these functions, and the generated image shows low consistency. our theoretical analysis shows that the gradients of the introduced two functions need to lie within two mutually orthogonal subspaces. Based on this analysis, we can design various metrics to compute the consistency for specific models. In contrast, RIGID focuses on the difference in gradient norm between natural and generated images for detection.
>
> **(3) Issues of implementation details:** We follow the data augmentation strategy used when training DINOv2 with a combination of HorizontalFlip, ColorJitter, and GaussianBlur. We've added the details to the revision.
>
> **(4) Issues of backbones:** Following your suggestion, we have implemented ConV on additional backbones. The experimental results demonstrate the universality of ConV.
>
> We understand that you're busy, but as the window for responding and revising the paper is closing, would you mind reviewing our response and confirming if you have any further questions? We would be happy to provide additional answers or revisions if needed.
>
> Best regards and thanks,
> Authors of #8640

---

### Official Review · Reviewer_t4Ze · 2024-11-04

**Soundness:** 2
**Presentation:** 3
**Contribution:** 3
**Rating:** 6
**Confidence:** 3

**Summary:**

This paper presents a framework, termed Consistency Verification (ConV), aimed at detecting AI-generated images. Different from traditional binary classifiers that require extensive collections of both natural and AI-generated images, ConV operates solely on the distribution of natural images. The authors introduce two functions whose outputs are consistent for natural images but highly different for AI-generated images.

**Strengths:**

- The underlying mathematical intuition of this paper is well-presented.
- The method looks novel to me and its effectiveness has been supported by a collection of experiments.

**Weaknesses:**

- **Theoretical Limitation**: Although the orthogonality principle is empirically validated, a formal proof of the convergence of ConV’s generalization risk is not provided, which could strengthen the theoretical grounding.

- **Clarity in Wording**: In line 135, the phrase, *"..., humans know that if a natural image $x_n$ captures the same content as $x_g$ , they are distinguishable in certain ways,"* could cause confusion. The authors might consider revising this to *"...even if a natural image $x_n$ captures similar content to $x_g$"* to clarify that subtle differences still make AI-generated and natural images distinguishable.

**Questions:**

As shown in Table 6, it appears that ConV is sensitive to the initialization function. ConV instantiated by CLIP performs much worse than DinoV2. Is there an approach to overcome this limitation?

---

> ### Author Response · Authors · 2024-11-23
>
> We would like to thank Reviewer t4Ze for taking the time to review our work. We appreciate your recognition of well-presented mathematical intuition, novelty, and effectiveness of our proposed method. Please see our detailed responses to your constructive comments below.
>
> > Weakness 1:
> A formal proof of the convergence of ConV’s generalization risk could strengthen the theoretical grounding.
>
> Response:
>
> We agree that a formal generalization risk could strengthen the theoretical contribution. It is challenging to develop a novel theory for a novel framework due to the limited time. Thus, we leave it as our future work.
>
> > Weakness 2: To avoid potential confusion, you should revise the description to clarify that subtle differences still make AI-generated and natural images distinguishable.
>
> Response:
>
> Thanks for your kind suggestion. We have revised it as follows:
>
> *In particular, for a given generated image $x_g$ , even if it captures similar content to a nature image $x_n$ , humans know they are distinguishable in certain ways.*
>
> > Weakness 3: As shown in Table 6, it appears that ConV is sensitive to the initialization function. ConV instantiated by CLIP performs much worse than DinoV2. Is there an approach to overcome this limitation?
>
> Response:
>
> We sincerely appreciate your insightful question. To overcome this limitation, we revisit our methodology, i.e., verifying the consistency between outputs of two functions.
>
> As shown in Eq. (13), we derive the cosine similarity metric between image features from a self-supervised learning objective function. However, CLIP employs a different objective function, namely, calculating the similarity between text and image features. Thus, the proposed cosine similarity between image features may not be a good realization of these two functions’ output, limiting the generalization capability for generated image detection.
>
> Inspired by your valuable question, we conjecture the difference between the projection of the visual features of the original image and the visual features of the transformed image on their corresponding text features would be a good metric. The reason is as follows.
>
> The function to calculate the similarity between text and image features can be regarded as function $f_1(x, t)$. Thus, we should calculate the difference in inter-modality similarity rather than the similarity between original and transformed images. To verify the point, we conduct experiments using the corrected realization of $f_1$ and $f_2$, i.e., a corrected metric to verify the consistency. The results below show that the modified approach outperforms the original metric.
>
> As shown in the following table, using the modified metric for detection greatly improves the performance of CLIP-based methods model, achieving performance comparable with Dinov2-based methods.
>
> We will incorporate the above analyses, results, and discussions into the revision. Inspired by your valuable question, we believe our work provides a novel approach to calculate the difference between two functions, without the focuses on the difference in similarities between two image features. Thanks again for your insightful and constructive question!
>
> |                                       | ADM   |       | ADMG  |       | Big GAN |       | DiT   |       | Giga GAN |       | LDM   |       | Mask-GIT |       | RQ-Transformer |       | StyleGAN-XL |       | Avg   |       |
> |---------------------------------------|-------|-------|-------|-------|---------|-------|-------|-------|----------|-------|-------|-------|----------|-------|----------------|-------|-------------|-------|-------|-------|
> |                                       | AUROC | AP    | AUROC | AP    | AUROC   | AP    | AUROC | AP    | AUROC    | AP    | AUROC | AP    | AUROC    | AP    |     AUROC        |  AP     |   AUROC       |  AP     | AUROC | AP    |
> | ConV (Dinov2)                         | 88.89 | 86.60 | 82.46 | 79.83 | 92.83   | 92.05 | 75.25 | 70.11 | 91.89    | 90.93 | 78.94 | 75.88 | 88.79    | 87.88 | 93.02          | 91.26 | 92.15       | 91.82 | 87.13 | 85.15 |
> | ConV (CLIP with image features)    | 76.64 | 76.52 | 69.36 | 68.86 | 76.59   | 79.27 | 70.03 | 69.73 | 72.97    | 73.05 | 70.29 | 69.73 | 72.95    | 73.20 | 77.27          | 77.49 | 70.82       | 70.35 | 72.99 | 72.98 |
> | ConV (CLIP with image and text features) | 80.76 | 79.77 | 72.31 | 71.21 | 80.73   | 76.60 | 72.73 | 72.12 | 79.59    | 77.47 | 72.03 | 71.22 | 74.34    | 70.45 | 80.83          | 78.86 | 77.46       | 75.17 | 76.75 | 74.76 |

---

> ### Author Response · Authors · 2024-11-27
>
> Dear Reviewer #t4Ze,
>
> Thank you very much for your time and valuable feedback.
>
> Here is a summary of our response for your convenience:
>
> **(1) Issues regarding ConV's generalization risk:** Developing a novel theory for a new framework is challenging due to the limited time available. We will address this in future work.
>
> **(2) Issues related to clarity in wording:** Following your suggestion, we have revised the sentence as follows:
>
> *In particular, for a given generated image $x_g$, even if it captures similar content to a nature image $x_n$, humans know they are distinguishable in certain ways.*
>
> **(3) Issues regarding ConV's performance on CLIP:** We have successfully improved the performance of ConV on CLIP by calculating the difference in inter-modality similarity, rather than the similarity between original and transformed images.
>
> We understand that you're busy, but as the window for responding and revising the paper is closing, would you mind reviewing our response and confirming if you have any further questions? We would be happy to provide additional answers or revisions if needed.
>
> Best regards and thanks,
> Authors of #8640

---

### Meta-Review · Area_Chair_gDrq · 2024-12-18

**Metareview:**

The recommendation is based on the reviewers' comments, the area chair's evaluation, and the author-reviewer discussion.

While the reviewers see some merits in deriving a statistic using a backbone model for consistency verification and the detection of AI-generated images, this submission should not be accepted in its current form due to several fundamental issues, as pointed out by the reviewers, including

- Technical novelty compared to existing methods, such as AeroBlade and RIGID. Notably, while te proposed framework is quite general, in terms of the actual implementation of Conv, it shares some similarities to RIGID (e.g., using DinoV2 as the backbone).
- Limited performance improvement: Compared to the baselines, the improvement is marginal and sometimes does not outperform the baselines.
- Theoretical contributions need to be reinforced.


During the final discussion phase, all reviewers agreed to reject this submission.

I hope the reviewers’ comments can help the authors prepare a better version of this submission.

**Additional Comments On Reviewer Discussion:**

This submission should not be accepted in its current form due to several fundamental issues, as pointed out by the reviewers, including

- Technical novelty compared to existing methods, such as AeroBlade and RIGID. Notably, while te proposed framework is quite general, in terms of the actual implementation of Conv, it shares some similarities to RIGID (e.g., using DinoV2 as the backbone).
- Limited performance improvement: Compared to the baselines, the improvement is marginal and sometimes does not outperform the baselines.
- Theoretical contributions need to be reinforced.


During the final discussion phase, all reviewers agreed to reject this submission.

I hope the reviewers’ comments can help the authors prepare a better version of this submission.

---

### Decision · Program_Chairs · 2025-01-22

Reject